# Exploring TCR-like CAR-Engineered Lymphocyte Cytotoxicity against MAGE-A4

**DOI:** 10.3390/ijms242015134

**Published:** 2023-10-13

**Authors:** Alaa Alsalloum, Julia Shevchenko, Marina Fisher, Julia Philippova, Roman Perik-Zavodskii, Olga Perik-Zavodskaia, Saleh Alrhmoun, Julia Lopatnikova, Kurilin Vasily, Marina Volynets, Evgenii Zavjalov, Olga Solovjeva, Yasushi Akahori, Hiroshi Shiku, Alexander Silkov, Sergey Sennikov

**Affiliations:** 1Laboratory of Molecular Immunology, Federal State Budgetary Scientific Institution Research Institute of Fundamental and Clinical Immunology, 630099 Novosibirsk, Russia; alaa.alsalloum19996@gmail.com (A.A.); shevchenkoja2023@yandex.ru (J.S.); msolshanova@gmail.com (M.F.); airyuka@mail.ru (J.P.); zavodskii.1448@gmail.com (R.P.-Z.); perik.zavodskaia@gmail.com (O.P.-Z.); saleh.alrhmoun1@gmail.com (S.A.); lopatnikova18@yandex.ru (J.L.); vkurilin@niikim.ru (K.V.); mrsmarinavolynets@gmail.com (M.V.); zavjalov@bionet.nsc.ru (E.Z.); solovieva@bionet.nsc.ru (O.S.); silkov@niikim.ru (A.S.); 2Faculty of Natural Sciences, Novosibirsk State University, 630090 Novosibirsk, Russia; 3Institute of Cytology and Genetics, Siberian Branch of the Russian Academy of Sciences, Ministry of Science and High Education of Russian Federation, 630090 Novosibirsk, Russia; 4Department of Personalized Cancer Immunotherapy, Mie University Graduate School of Medicine, Tsu 514-8507, Mie, Japan; yakahori@med.mie-u.ac.jp; 5Department of Immunology, V. Zelman Institute for Medicine and Psychology, Novosibirsk State University, 630090 Novosibirsk, Russia

**Keywords:** TCR-like CAR-T cells, MAGE-A4, GITR, cytotoxicity, Nanostring

## Abstract

TCR-like chimeric antigen receptor (CAR-T) cell therapy has emerged as a game-changing strategy in cancer immunotherapy, offering a broad spectrum of potential antigen targets, particularly in solid tumors containing intracellular antigens. In this study, we investigated the cytotoxicity and functional attributes of in vitro-generated T-lymphocytes, engineered with a TCR-like CAR receptor precisely targeting the cancer testis antigen MAGE-A4. Through viral transduction, T-cells were genetically modified to express the TCR-like CAR receptor and co-cultured with MAGE-A4-expressing tumor cells. Flow cytometry analysis revealed a significant surge in cells expressing activation markers CD69, CD107a, and FasL upon encountering tumor cells, indicating robust T-cell activation and cytotoxicity. Moreover, immune transcriptome profiling unveiled heightened expression of pivotal T-effector genes involved in immune response and cell proliferation regulation. Additionally, multiplex assays also revealed increased cytokine production and cytotoxicity driven by granzymes and soluble Fas ligand (sFasL), suggesting enhanced anti-tumor immune responses. Preliminary in vivo investigations revealed a significant deceleration in tumor growth, highlighting the therapeutic potential of these TCR-like CAR-T cells. Further investigations are warranted to validate these revelations fully and harness the complete potential of TCR-like CAR-T cells in overcoming cancer’s resilient defenses.

## 1. Introduction

In recent years, significant strides have been made in the development of personalized treatment methods tailored specifically to address the needs of individuals afflicted with malignant neoplasms [1]. The rapid advancements in this scientific field have led to the emergence and meticulous evaluation of a diverse range of genetically modified lymphocytes. Notably, within this repertoire of lymphocytes, T-cells have been equipped with chimeric antigen receptors (CAR-T cells), intricately engineered to selectively recognize surface molecules present on tumor cells. Additionally, T-cells have been expertly engineered to incorporate artificially introduced T-cell receptors (TCR-engineered T-cells), granting them the precise specificity desired for effective therapeutic interventions [2,3,4].

CAR-T lymphocytes recognize surface antigen epitopes, which comprise 14–26% of the total cell proteome [5], limiting the number of potential effective and specific targets for anticancer CAR-T therapy. However, the intracellular domain of the chimeric receptor can be modified to transmit extra costimulatory signals that activate T-cells, ensuring resistance to the immunosuppressive tumor microenvironment and depletion [6,7,8].

TCR-engineered T-cells, similar to natural T-lymphocytes, recognize the peptide-MHC complex (pMHC), which can be derived from antigens from any cellular compartment, including neoantigens, antigens of oncogenic viruses, and germline cancer antigens (cancer-testis antigens) [9]. At the same time, proper activation of TCR-engineered T-cells requires a full-fledged co-stimulatory signal, which can be challenging to implement in an immunosuppressive tumor microenvironment. However, the efficiency of anticancer therapy can be improved by following immunization with pMHC-carrying agents such as MHC monomers or antigen-presenting cells [10].

A potential approach to combining the advantages and avoiding the disadvantages of CAR-T-cell and TCR-engineered cell therapy can be through the application of lymphocytes with a TCR-like CAR-receptor [11,12]. In this scenario, the antigen-recognition domain is composed of a single-chain variable fragment (scFv) derived from a TCR-like antibody that specifically targets the desired epitope-MHC complex. Meanwhile, the intracellular signaling domain of these lymphocytes is similar to that of CAR-T cells. Consequently, lymphocytes with a TCR-like CAR receptor can be directed to antigens at any cellular location [13]. Modifications to the CAR receptor’s intracellular signaling domain, as well as vaccination with pMHC-carrying agents, may promote its function.

This paper describes the generation, as well as the phenotypic, and functional properties of lymphocytes with a TCR-like CAR receptor, the antigen-recognizing region of which is specific to the p230–239 epitope (GVYDGREHTV) of the tumor antigen MAGE-A4 (melanoma-associated antigen family A4), and the intracellular domain contains the activation motif GITR (glucocorticoid-induced TNFR-related protein, TNFRSF18, CD357).

MAGE-A4 levels were found to be significantly higher in malignant neoplasms of the head and neck, esophagus, stomach, ovaries, endometrial, and other organs [14,15,16]. MAGE-A4 is a cancer-embryonic antigen that is primarily expressed in immune-privileged tissues such as the testicles and placenta. This makes it a promising candidate for targeted antitumor therapy with low toxicity. Previous efforts to target MAGE-A4 using protein and peptide vaccines, as well as TCR-engineered lymphocytes, have demonstrated limited clinical effectiveness [17]. Therefore, the quest for alternative approaches to anti-MAGE-A4 therapy remains relevant.

We assumed that the inclusion of the GITR activation motif within the intracellular domain of the CAR construct in the proposed approach holds the potential to improve its therapeutic efficacy. This enhancement can be attributed to the robust T-cell costimulatory activity exerted by GITR [18,19]. Moreover, the intracellular signaling mediated by GITR, resulting in NF-kB activation, remains unimpeded by the inhibitory effects of PD-1 signaling [20].

In our work, we employed in vitro transduction with viral vectors to generate TCR-like CAR-T cells. We then conducted an extensive investigation to evaluate the phenotype, activation status, and cytotoxic potential of these TCR-like CAR-T cells upon exposure to MAGEA4-positive tumor cell lines. Additionally, we assessed the therapeutic potential of these cells using an in vivo melanoma model. The findings in this study may provide substantial insights for the advancement of anticancer treatment through the utilization of genetically modified lymphocytes.

## 2. Results

### 2.1. Anti-MAGE-A4 TCR-like CAR-T Cells Manifest Cytotoxicity Markers

We conducted an in-depth analysis of flow cytometry data from T-lymphocyte cells using HSNE dimensionality reduction. An analysis was conducted on the transduced T-cells after co-cultivating them with the SK-Mel-37 cell line expressing MAGE-A4, with the aim of accurately classifying cells based on their marker expression. The clustering analysis unveiled distinct subpopulations of T-cells. Notably, the predominant subpopulation consisted of CD8^+^ CD69^+^ CD107a^+^ FasL^+^ CD40L^−^ 4-1BB^−^ lymphocytes, comprising approximately 68.47 ± 3.041% of the T-cells. Additionally, CD4^+^ FasL^+^ CD69^+^ CD107a^+^ CD40L^−^ 4-1BB^−^ cells constituted 26.12 ± 3.780%, while CD8^+^ CD69^+^ CD107a^+^ FasL^+^ CD40L^−^ 4-1BB^−^ lymphocytes accounted for 5.411 ± 1.167% (mean ± standard error of the mean, *n* = 6) (Figure 1 and Figure 2).

### 2.2. Transcriptional Profiling Reveals Immunological Features of Generated TCR-like CAR-T Cells

We then conducted differential gene expression analysis of the transduced T-cells’ immune transcriptome before and after the co-cultivation with the SK-Mel-37 cell line expressing MAGE-A4. We considered *q*-values < 0.005 and log2 (Fold change) > 0.847 or log2 (Fold change) < −0.847 significant (Figure 3). 

The analysis revealed a set of genes with altered expression levels. Among the up-regulated genes were: *AHR*, *APP*, *ATG12*, *ATG7*, *BCL6*, *BST1*, *C1QBP*, *C1R*, *C1S*, *C3*, *CCL2*, *CCL20*, *CCL3*, *CCL4*, *CD22*, *CD276*, *CD59*, *CD9*, *CEBPB*, *CFI*, *CIITA*, *CSF1*, *CSF2*, *CTNNB1*, *CXCL1*, *CXCL10*, *CXCL11*, *CXCL2*, *EDNRB*, *EGR1*, *EGR2*, *FCGR2A*, *FN1*, *GZMB*, *HLA-DMA*, *HLA-DMB*, *HLA-DPA1*, *HLA-DPB1*, *HLA-DRA*, *HLA-DRB1*, *HLA-DRB3*, *ICAM1*, *IFNG*, *IL13*, *IL13RA1*, *IL1A*, *IL1B*, *IL1RAP*, *IL2*, *IL22*, *IL3*, *IL6*, *IL6ST*, *IL8*, *IRAK2*, *ITGA6*, *LIF*, *LTBR*, *NCAM1*, *NT5E*, *PLAU*, *PLAUR*, *PRKCD*, *PSMB5*, *PSMB7*, *PTK2*, *SMAD5*, *SOCS3*, *SPP1, TBX21*, *TGFBI*, *THY1*, *TLR2*, *TNFRSF9*.

Conversely, the down-regulated genes comprised: *ADA*, *ARHGDIB*, *B2M*, *BAX*, *BCL2*, *CASP1*, *CASP8*, *CCL5*, *CCND3*, *CCR2*, *CCR5*, *CD244*, *CD247*, *CD27*, *CD28*, *CD3D*, *CD3E*, *CD45RB*, *CD5*, *CD53*, *CD6*, *CD7*, *CD79B*, *CD8A*, *CD8B*, *CD96*, *CD99*, *CISH*, *CSF2RB*, *CX3CR1*, *CXCR4*, *DPP4*, *FOXP3*, *GBP5*, *GZMA*, *HLA-A*, *HLA-B*, *ICAM2*, *ICAM3*, *ICOS*, *IFITM1*, *IKBKE*, *IKZF1*, *IL10RA*, *IL12RB1*, *IL16*, *IL2RG*, *IL4R*, *IRAK4*, *ITGAL*, *ITGAM*, *ITGB2*, *JAK1*, *JAK2*, *JAK3*, *KLRC4*, *LAIR1*, *LCK*, *LCP2*, *LEF1*, *MAP4K1*, *MR1*, *MUC1*, *MYD88*, *NCF4*, *NFATC3*, *PDGFRB*, *POU2F2*, *PRDM1*, *PRF1*, *PSMB10*, *PTGER4*, *PYCARD*, *RARRES3*, *S1PR1*, *SELL*, *SELPLG*, *SIGIRR*, *SLAMF1*, *SLAMF6*, *STAT2*, *STAT4*, *STAT5B*, *TAP1*, *TAP2*, *TGFB1*, *TGFBR1*, *TLR1*, *TMEM173*, *TNFSF12*, *TNFSF8*, *UBE2L3*, *ZAP70*.

After that, we conducted gene set enrichment analysis of the up-regulated genes in gene ontology biological process terms (Figure 4, Table 1).

### 2.3. TCR-like CAR-T Cells Exhibit Antigen-Dependent Activation and Cytotoxicity In Vitro

We proceeded to investigate the antigen-dependent activation and cytotoxic potential of transduced T-lymphocytes. The transduced T-cells, acting as effectors, were co-cultured with human tumor cell lines SK-Mel-37, NW-Mel-38, and HCT-116 for a duration of 6–8 h, maintaining an effector-to-target ratio of 5:1. Our analysis revealed a significant increase in the cytotoxic response against cell lines expressing the target antigen in comparison to the MAGEA4 negative line. Interestingly, there were no notable differences observed in the cytotoxicity levels between the NW-MEL-38 and SK-MEL-37 cell lines (Figure 5).

### 2.4. TCR-like CAR-T Cells Elicit the Secretion of Key Cytokines in Response to Tumor Cells

We conducted a comprehensive analysis of cytokine secretion in the conditioned media from two groups: transduced cells and non-transduced cells, both cultured with SK-Mel-37 tumor cells. The results of our analysis demonstrated significant differences in the secretion levels of pivotal cytokines: granzyme B, IFN-gamma, TNF-alpha, sFasL, and IL-2 (Figure 6, Table 2). Notably, the transduced cells exhibited substantially higher secretion levels of these cytokines compared to the non-transduced cells in response to the SK-Mel-37 tumor cells.

### 2.5. Anti-MAGEA4 TCR-like CAR Slows Tumor Growth in Melanoma Model In Vivo

To evaluate the efficacy of MAGEA4-directed TCR-like CAR-T cells in a solid tumor model in vivo, we introduced human melanoma cells SK-Mel-37 expressing MAGEA4 into NRG mice. Once the average tumor volume reached 100 mm^3^, we randomly assigned the mice into three groups (untreated, control, experimental), each consisting of eight mice. In the control group, we administered non-transduced T-cells, while in the experimental group, we treated the mice with anti-MAGE-A4 TCR-like CAR-T cells. Both treated groups received a single dose of 8 million cells per mouse, while the untreated group did not receive any treatment. Starting from day 22, mice treated with Anti-MAGE-A4 TCR-like CAR-T cells exhibited a significant reduction in tumor growth compared to the untreated group. Furthermore, a single dose of anti-MAGE-A4 TCR-like CAR-T cells significantly slowed tumor growth compared to mice in the control group, starting from day 32 (Figure 7).

## 3. Discussion

In this study, we embarked on a comprehensive exploration of the intricate mechanisms involved when a TCR-like CAR (chimeric antigen receptor) binds to a pMHC (peptide–major histocompatibility complex) on the surface of tumor cells. This binding event triggers a cascade of intracellular signaling events within the TCR-like CAR-T cells, ultimately activating them to seek out and eliminate tumor cells. Importantly, this TCR-like CAR system mimics the functionality of natural T-cell receptors (TCRs) but offers distinct advantages over traditional transgenic TCRs [21]. This novel receptor design, akin to affinity-enhanced TCRs, holds the potential to revolutionize T-cell-based immunotherapies, offering improved antigen specificity and recognition capabilities [22,23].

Our investigation was centered on analyzing flow cytometry data, employing HSNE (hierarchical stochastic neighbor embedding) dimensionality reduction, to delve into the dynamic interplay of immune responses. Specifically, we focused on T-cell interactions with the SK-Mel-37 cell line expressing MAGE-A4, an antigen of interest in melanoma immunotherapy. The outcomes of our analysis unveiled a diverse landscape of T-cell subpopulations, each defined by unique marker expression profiles.

One particularly intriguing T-cell subpopulation stood out: CD8^+^ CD69^+^ CD107a^+^ FasL^+^ CD40L^−^ 4-1BB^−^ lymphocytes. This subset exhibited characteristics strongly associated with cytotoxicity and the targeted eradication of tumor cells. The presence of CD107a granular membrane proteins reinforced the notion that these T-cells play a pivotal role in eliminating tumor cells [24]. Equally noteworthy was the expression of Fas ligand (FasL), hinting at a potential mechanism by which these T-cells induce apoptosis in target tumor cells [25]. Adding another layer of complexity to our findings, we identified CD4^+^ FasL^+^ CD69^+^ CD107a^+^ CD40L^−^ 4-1BB^−^ cells, suggesting that CD4^+^ T-cells might also participate in cytotoxic functions. Traditionally, CD8^+^ T-cells have held the spotlight as the primary effectors of cytotoxic responses. However, emerging research points to the capacity of CD4^+^ T-cells to exhibit cytotoxic capabilities under specific conditions [26], opening new avenues for our understanding of anti-tumor immunity.

Moreover, the early activation marker CD69 signals that these T-cell subpopulations swiftly responded upon encountering MAGE-A4-expressing tumor cells [27].

Beyond the phenotypic characterization of T-cell subsets, our research extended to a comparative analysis of the immune transcriptome. We scrutinized gene expression in transduced T-cells both before and after co-cultivation with MAGE-A4-expressing SK-Mel-37 cells. This comprehensive analysis, coupled with gene set enrichment analysis (GSEA) of up-regulated genes, offered compelling evidence of immune function activation. We observed the up-regulation of pivotal immune-related genes known for their cytotoxic effects, including *GZMB* (granzyme B) and *IFNG* (interferon-gamma). This suggests a robust immune response capable of enhancing the anti-tumor potential of transduced T-cells [28].

One notable observation was the significant upregulation of the *FCGR2A* gene, which encodes CD32 receptors. These receptors are crucial for binding to the Fc portion of IgG, highlighting their essential role in bridging innate and adaptive immunity. Interestingly, our research hinted at the potential of CD4^+^ T-cells to influence the expression of CD32 receptors, both on the cell surface and intracellularly [29].

Furthermore, our investigation unveiled an increased transcriptional activity of genes encoding chemokines. These genes play established roles in directing T-cell trafficking, facilitating infiltration into tumor microenvironments, governing migratory patterns, and orchestrating intricate cellular interactions [30]. Some of these chemokines could potentially emerge as significant products of activated CD4^+^ lymphocytes, akin to traditional cytokines. Notably, the interaction between CD4^+^ T-cells and antigen-presenting dendritic cells (DCs) was found to induce the production of CCL3 and CCL4 chemokines, which bind to CCR5 on naive CD8^+^ T-cells, promoting their migration to sites of immune activation [31].

One of the central players highlighted in our research was *IL-2* (Interleukin-2), known for its potent mitogenic and growth-inducing properties. Extensive prior studies have primarily explored IL-2 signaling in antigen receptor-activated CD4^+^ and CD8^+^ T-cells. IL-2 plays a pivotal role in precisely regulating cytokine receptors, transcription factors, chromatin regulators, and effector cytokines, significantly influencing T-cell fate. Among these factors, T-bet, encoded by *Tbx21*, emerges as a critical regulator [32]. It plays a multifaceted role in the differentiation and effector functions of cell-mediated immunity. Specifically, T-bet’s influence extends to T helper type 1 (Th1) cell differentiation, particularly through its regulation of interferon-gamma (IFNG) expression. This regulatory role extends to cytotoxic CD8^+^ T-cells and natural killer (NK) cells, impacting the expression of IFN-gamma, perforin, and granzyme B [33,34,35]. Furthermore, collaborative mechanisms, such as the interplay between T-bet and Bcl-6 in regulating Th1 gene expression patterns, contribute to maintaining IFNG expression within a balanced range [36].

Our transcriptome analysis may indicate the coexistence of effector differentiation and memory development within a subset of cells, consistent with previous research. Importantly, Bcl6 has been closely linked to the generation and proliferation of memory CD8^+^ T-cells [37,38]. This promising characteristic could potentially enhance both rapid proliferation and sustained immune responses in vivo. However, further investigations are warranted, particularly concerning the metabolic aspects of this regulation, offering the prospect of deeper insights into the realms of immunology and immunotherapy.

Notably, our research showcased a down-regulation of crucial regulatory T-cell (Treg) function genes, including *FOXP3*, *TGFB1*, and *TGFBR1* [39,40]. This observation implies a potential reduction in the immunosuppressive microenvironment within the T-cell response. Complementing these findings, our comprehensive analysis of various cytokine proteins indicated no significant secretion levels of the anti-inflammatory cytokines IL-10 and IL-4 (Figure 6) [41,42]. These observations could be attributed to the co-stimulatory GITR signaling previously mentioned [18]. The upregulation of GITR on effector T-cells upon TCR activation has been shown to inhibit the suppressive activity of Tregs and enhance the survival of effector T-cells [43]. This modulation of the immune landscape potentially creates a more favorable environment for CAR-T-cell activity, further augmenting their anti-tumor effects.

Intriguingly, our investigation into gene expression at the protein level has unveiled increased production of granzyme B, IFN-gamma, TNF-alpha, and soluble Fas ligand (sFasL) pin-transduced TCR-like CAR-T cells upon encountering SK-Mel-37 tumor cells. These findings are consistent with the established mechanism of CAR-T-cell therapy [28], suggesting that the cytotoxic activity of TCR-like CAR-T cells involves a combination of degranulation and ligand-based lytic pathways, which may work synergistically or additively. Particularly, it is interesting to note that FasL can enhance lytic action even in instances of poor or hampered degranulation [44,45].

Furthermore, the investigation of the immune transcriptome revealed a down-regulation in the expression of the perforin gene (*PRF1*). This was supported by cytokine analysis (Figure 6), which indicated no notable production of perforin. These findings point to a possible mechanism in which granzyme B-mediated activity is independent of perforin. However, the precise molecular mechanisms underlying granzyme delivery remain unclear, as conflicting reports have emerged, leading to disparate findings in the scientific literature [46].

It is essential to acknowledge the potential translational and clinical implications of our findings. Although our in vitro approach provided valuable insights, it is vital to consider its limitations, particularly in assessing potential off-target effects and toxicity. Previous studies have indicated that single-chain variable fragments (scFvs), such as those used in CARs, can exhibit cross-reactivity with different peptides, recognizing unique conformations of MHC proteins bound to specific peptides or directly binding to peptides, akin to T-cell receptors (TCRs) [47,48]. Furthermore, the release of cytokines by activated T-cells during treatment can lead to immune-related adverse events such as cytokine release syndrome (CRS) and neurotoxicity [49].

To bridge the gap between preclinical research and clinical translation, we conducted in vivo studies using immunodeficient mice bearing melanoma tumors. Our preliminary and preclinical findings demonstrated a significant reduction in tumor growth in the experimental group compared to both the untreated and control groups. Although complete tumor regression was not attained, the observed reduction in tumor growth is particularly promising. This is noteworthy, especially considering that only a single dose of anti-MAGE-A4 TCR-like CAR-T cells was administered in this study. Given the well-documented dose–response relationship in CART-cell therapy, the optimization of treatment strategies through precise dosage adjustments, the exploration of synergistic combination therapies, and the implementation of multiple dosages exhibit significant potential for enhancing treatment efficacy [50,51].

In summary, our study provides a comprehensive understanding of TCR-like CAR T-cell responses, offering promising insights into improving immunotherapy strategies for cancer treatment.

## 4. Materials and Methods

### 4.1. Study Population and Interventions

Venous blood sampling was conducted on a cohort of six healthy adult donors. The average age of the healthy blood donors was determined to be 27.33 ± 6.34 years (mean ± standard error of the mean).

### 4.2. PBMC Isolation

We performed venous blood collection, drawing up to 5 mL of blood, and preserved the samples in EDTA-containing tubes. We isolated peripheral blood mononuclear cells (PBMCs) from whole blood samples using a conventional Ficoll–Urografin (PanEco, Moscow, Russia) density gradient method. Briefly, we diluted the peripheral blood with an equal volume of RPMI-1640 medium (Biolot, St. Petersburg, Russia). The diluted blood was then layered on top of a Ficoll–Urografin solution (ρ = 1.077 g/L) and subjected to centrifugation at 400× *g* and room temperature for 40 min. Subsequently, the mononuclear cells were harvested from an opalescent layer located at the phase boundary, spanning the entire cross-section of the tube.

### 4.3. Retroviral Particles

We were provided with a gamma-retroviral vector by Prof. H. Shiku from Mie University Graduate School of Medicine, Japan. This vector contains an insert encoding a specialized T-cell receptor (CAR) designed exclusively to target the p230–239 epitope (GVYDGREHTV) of the tumor antigen MAGE-A4 (Figure 1).

### 4.4. Induction of T-Cell Proliferation

We stimulated the proliferation of peripheral blood mononuclear cells (PBMCs) by immobilizing retronectin at a concentration of 25 mg/mL (Takara Bio, Kusatsu, Japan) and CD3 antibodies at a dose of 5 mg/mL (Biolegend, San Diego, CA, USA) onto the wells of 12-well plates (TPP, Trasadingen, Switzerland). The PBMCs were cultured at a concentration of 0.5 × 10^6^ cells/mL in 2 mL of GT-T551 medium (Takara Bio, Japan) supplemented with 300 U/mL IL-2 (Roncoleukin, Biotech, Moscow, Russia) and 0.6% human blood serum of group AB. The plates were then incubated in a humidified atmosphere with 5% CO_2_ at a temperature of 37 °C. During the 2nd and 3rd days of cultivation, we actively replenished half of the culture medium volume with a fresh portion of IL-2.

### 4.5. Retroviral Transduction of the Anti-CD3 Primed PBMCs

To perform retroviral transduction, we thawed 1 mL of a retrovirus solution in a water bath at 37 °C. Then, we diluted it four times in PBS (Biolot, St. Petersburg, Russia) with 2% human serum albumin (Microgen, Tomsk, Russia), and a 5% glucose-citrate buffer. We added the diluted retroviral solution to retronectin-coated wells of 24-well plates and centrifuged the plate with the retrovirus solution for 2 h at 32 °C at 2000× *g* (Jouan MR 23, Nantes, France). After the liquid content was removed, the wells were washed twice with PBS containing 2% albumin. Subsequently, 1.5–2 × 10^5^ PBMCs primed with anti-CD3 antibodies were added to each well, resuspending them in 1.5 mL of medium supplemented with IL-2 (300 U/mL), and the cell suspension plates were centrifuged at 1000× *g* for 10 min at 32 °C. The cell plates were placed in a humidified atmosphere with 5% CO_2_ at 37 °C. The next day, a second round of transduction was performed by transferring the cells into the wells of new 24-well plates with retroviral particles. The cells were then centrifuged at 1000× *g* for 10 min at 32 °C, which allowed for an increase in the efficiency of transduction of 10–15%. After that, the cells were incubated in a humid atmosphere at 37 °C containing CO_2_. Following 6–8 h, the cells were transferred to new 6-well plates (TPP, Trasadingen, Switzerland) supplemented with an equivalent volume of GT-T551 culture medium and 300 U/mL IL-2. On days 9–10 after initiating the protocol, an aliquot of cells was collected to assess transduction efficiency and determine the cell phenotype. On the 11th day from the start of the protocol, the transduced cells were co-cultured with tumor cell lines to evaluate their cytotoxic activity in vitro. As a control, we used cells primed with anti-CD3 and stimulated with IL-2, to which no retrovirus was added, and no transduction was performed.

### 4.6. Evaluation of the Efficiency of Transduction

We employed biotin-SP (long spacer) AffiniPure F(ab′)2 Fragment Goat Anti-Mouse IgG, F(ab′)2 fragment specific (Jackson Immunoresearch, West Grove, PA, USA) to assess the transduction efficiency of PBMCs. The cells were stained with these antibodies at a 1:1000 dilution and incubated for 20 min in the dark at room temperature. After two washes, we added streptavidin-PE at a dilution of 1:600, along with monoclonal antibodies targeting surface markers (anti-CD3-AF700 (Biolegend, USA)) and Zombie Aqua vital dye (Biolegend, USA) to evaluate the culture’s viability. Staining was conducted for 20 min at room temperature in the dark. The stained cells were then washed with PBS containing 0.1% NaN3 and subjected to analysis using an Attune NxT flow cytometer (Thermo Fisher, Waltham, MA, USA). We gated cells from debris, singlets from the cells, alive cells from the singlets, CD3-positive cells from the alive cells, and anti-Fab-positive cells from the CD3-positive cells. We observed 50 ± 12.28% retroviral transduction of said T-cells (mean ± standard deviation), with a minimum of 37.7% and a maximum of 62.2% (Figure 2).

### 4.7. Cell Lines

We utilized human melanoma cell lines NW-Mel-38 and SK-Mel-37 as cell lines that express the MAGEA4 target antigen. For our experiments, we used the HCT-116 cell line as a negative control, which does not express the MAGE-A4 target antigen. The tumor cell lines were cultivated in culture flasks with a planting concentration of 50–75 thousand cells/mL until they reached the log-phase of growth. RPMI-1640 medium supplemented with 10% FCS (Hyclone, Logan, UT, USA), 2 mM L-glutamine (Biolot, Russia), 5 × 10^−5^ mM mercaptoethanol (Sigma, Cibolo, TX, USA), 25 mM HEPES (Sigma, USA), 80 µg/mL gentamicin (Krka, Novo Mesto, Slovenia), and 100 µg/mL ampicillin (Sintez, Kurgan, Russia) were used for cell culture. After detachment from the plastic surface using trypsin-versen solution (Biolot, St. Petersburg, Russia), the cells were seeded into the wells (4–5 × 10^4^ cells/well) of a 96-well flat-bottomed culture plate (TPP, Switzerland) for 16–17 h, ensuring proper adhesion of the tumor cells to the plastic. The cell lines NW-Mel-38 and SK-Mel-37 were obtained from the collection of cell cultures at the Graduate School of Medicine, Mie University, under the supervision of Professor H. Shiku.

### 4.8. Phenotyping for Markers of Activation and Cytotoxicity by Flow Cytometry

We aimed to explore the antigen-dependent activation level and cytotoxic potential of transduced T-lymphocytes by co-culturing the studied T-cells with tumor target cells. The adherent tumor cells (target) were initially cultured for 16–17 h, after which they were replaced with fresh culture medium, and T-cells (effectors) were introduced to achieve an effector: target ratio of 5:1. We co-cultured the transduced T-cells with human tumor lines SK-MEL-37, NW-Mel-38, and HCT-116 for 4–5 h. We conducted an analysis of activation and cytotoxicity marker expression using flow cytometry, employing a range of monoclonal antibodies, including anti-CD3-AF700 (cat #300324), anti-CD8-PeCy7 (cat #344712), anti-CD4-Bv570 (cat #300534), CD137(4-1BB)-BV711TM (cat #309832), anti-CD154 (CD40L)-PerCP/Cy5 (cat #310834), anti-CD69-AF647 (cat #310918), anti-CD178 (FasL)-BV421TM (cat #306412), and anti-CD107a-APC/Cyanine7 (cat #328630). All antibodies were sourced from Biolegend (USA) and used in strict accordance with the manufacturer’s instructions. For MAGEA4-specific cell labeling, we utilized anti-Fab as previously described. Following the staining process, the cells were washed with PBS containing 0.1% NaN3 and subsequently analyzed using an Attune NxT flow cytometer (Thermo Fisher, USA). We then gated cells from debris, singlets from the cells, alive cells from the singlets, and CD3-positive cells from the alive cells, and exported them as.fsc files in the conventional gating software for the Attune NxT Flow cytometer. We then transformed the .fcs files to .csv files using a custom Python 3 code via Jupyter Notebooks. We performed arcsinh-transformation with the automatically selected cofactors of the flow cytometry data contained in the .csv files to automatically split cells into negative and positive for all the markers with the R script (https://github.com/janinemelsen/Single-cell-analysis-flow-cytometry/blob/master/scripts/CSV_to_transformed_normalized_FCS_git.R, accessed on 16 July 2023) published by Melsen et al. [52]. We then performed simultaneous batch correction and data normalization by fdaNorm and exported the corrected .csv files as .fcs with the R script (https://github.com/janinemelsen/Single-cell-analysis-flow-cytometry/blob/master/scripts/clustering_dimensionalityreduction_pseudotime_git.R, accessed on 16 July 2023) originally published by Melsen et al. [52].

### 4.9. HSNE Dimensionality Reduction and Clustering

We normalized the .fcs files into the Cytosplore app [53] and subjected them to an HSNE dimensionality reduction. We used CD4, CD8, CD40L, CD69, CD107a, CD137 (4-1BB), and FasL (CD178) for dimensionality reduction. We then exported the frequencies of the cells per cluster and visualized them in GraphPad Prism 9.4 as box plots.

### 4.10. Magnetic Separation of Transduced T-Cells after Co-Culturing with Tumor Cells

To investigate the gene expression profile, we isolated MAGEA4-positive cells from the overall cell population following retroviral transduction. We accomplished this by adding biotin-SP (long spacer) AffiniPure F(ab′)2 Fragment Goat Anti-Mouse IgG, F(ab′)2 fragment-specific antibodies to the cells on day 11 of the experiment (10 µL of antibodies per 10 × 10^6^ cells), followed by a 20 min incubation in a cold Versene solution supplemented with 0.5% bovine serum albumin. After two washes, we added magnetic beads conjugated with MojoSort™ Streptavidin Nanobeads (Biolegend, USA, cat #480016) to the cells at a rate of 10 µL of beads per 10 × 10^6^ cells. We then washed and sorted the cells using a MojoSort™ Magnet (Biolegend, USA, cat #480019). The sorted cells were allowed to rest for 16–17 h in a culture medium supplemented with IL-2 (300 U/mL). Next, we co-cultured these cells with adherent MAGEA4-positive tumor cells (SK-Mel-37) at an effector-to-target ratio of 5:1, maintaining the co-culture for 2 h to activate key genes that regulate the immune response. Following the co-cultivation with tumor cells, we separated the transduced T-cells from the tumor cells using intensive pipetting in PBS. To eliminate any tumor cell contamination, we conducted positive magnetic sorting for the MojoSort™ Human CD45 Nanobeads (Biolegend, USA, cat #480029) marker. The viability of the magnetically sorted cells was then assessed using a Countess 3 Automated Cell Counter (Thermo Fisher Scientific, Waltham, MA, USA) and trypan blue staining, revealing a cell viability of over 92%.

### 4.11. Total RNA Extraction

We isolated total RNA from 300.000 to 600.00 cells with the Total RNA Purification Plus Kit (Norgen Biotek, Thorold, ON, Canada). We measured the concentration and quality of the total RNA in each sample on a Nanodrop 2000 spectrophotometer (Thermo Fisher Scientific, USA). We froze the total RNA at −80 °C until analysis.

### 4.12. Gene Expression Profiling by Nanostring

We performed gene expression profiling with the help of the Nanostring nCounter SPRINT Profiler analytical system using 100 ng of total RNA from each sample. We used an nCounter Human Immunology v2 panel to analyze the total RNA samples. The nCounter Human Immunology v2 panel consists of 579 immune and inflammation-associated genes, 15 housekeeping genes, and eight negative and six positive controls. The samples (*n* = 3–6) were subjected to an overnight hybridization reaction at 65 °C, where 5–14 μL of total RNA was combined with 3 μL of nCounter Reporter probes, 0–7 μL of DEPC-treated water, 11 μL of hybridization buffer, and with 5 μL of nCounter capture probes (total reaction volume = 33 μL). After the hybridization of the probes to targets of interest in the samples, the number of target molecules was determined on the nCounter digital analyzer. We performed normalization and QC in nSolver 4 using added synthetic positive controls and the 15 housekeeping genes included in the panel. We then performed background thresholding on the normalized data to remove non-expressing genes. The background level was determined as: mean of all NEG controls + 2SD of all NEG controls + mean of the POS_E controls.

### 4.13. Differential Gene Expression Testing

We performed differential gene expression using multiple *t*-tests (with Q < 0.005) in GraphPad Prism 9.4. The Volcano plot was created in GraphPad Prism 9.4. The GSEA was done using GSEApy [54].

### 4.14. LDH Cytotoxicity Assay

After allowing the adherent tumor cells (target) to incubate for 16–17 h, we replaced the culture medium with X-VIVO 15 serum-free medium (Lonza) and then added T-cells (effectors) to achieve an effector: the target ratio of 5:1 Cytotoxicity was assessed by determining the lactate dehydrogenase (LDH) activity in the conditioned medium of transduced T-cells and human tumor lines SK-MEL-37, NW-Mel-38, and HCT-116 for a duration of for 6–8 h, using the CytoTox 96 Non-Radioactive Cytotoxicity Assay (G1780, Promega Corporation, Madison, WI, USA). LDH activity was measured through a 30 min coupled enzymatic assay, which converts the tetrazolium salt INT into red formazan. The absorbance of visible light was determined using a standard 96-well plate reader (Varioskan, Thermo Fisher Scientific), and the intensity of the color formed was found to be directly proportional to the number of lysed cells. For statistical analysis of LDH cytotoxicity, we employed one-way ANOVA with Dunnett correction for multiple testing (with *q*-value < 0.0001) using GraphPad Prism 9.4.

### 4.15. Quantification of Cytokine Production

We co-cultured transduced cells with SK-Mel-37 tumor cells at an effector: target ratio of 5:1 for 48 h. Co-culture supernatants were collected and immediately frozen until the time of analysis. We employed the LEGENDplex™ Human CD8/NK Panel (13-plex) with a Filter Plate kit (Cat. Number: 740267, Lot Number: B330268, Biolegend, USA) to assess the cytokine content in the conditioned media. The analysis was performed following the manufacturer’s instructions. For each sample, 25 µL of conditioned medium was used. Subsequently, we compared the resulting cytokine profiles for transduced cells cultured with SK-Mel-37 tumor cells to those of non-transduced cells cultured with SK-Mel-37 tumor cells, aiming to identify any differences in cytokine production between the two experimental conditions. Statistical analysis was conducted using GraphPad Prism 10.0.0 software (GraphPad Software, San Diego, CA, USA). The Mann–Whitney test was used to compare between the two groups. Cytokines were categorized based on their respective concentration levels. The data are presented as median and the interquartile range.

### 4.16. In Vivo Efficacy

NRG immunodeficient mice of the NOD.Cg-Rag1tm1Mom Il2rgtm1Wjl/SzJ strain were utilized. The study was conducted at the Center for Genetic Resources of Laboratory Animals within the Institute of Cytology and Genetics, Siberian Branch of the Russian Academy of Sciences (RFMEFI62119X0023). Both male and female mice, aged 8 weeks, were included in the study, all with SPF (specific pathogen-free) status. These mice were housed in single-sex family groups, comprising 2–5 individuals per group, within individually ventilated cages (IVC) using the Opti Mice system provided by Animal Care Systems. These cages maintained controlled environmental conditions, including a temperature range of 21–24 °C, relative humidity levels between 30–50%, and a lighting regimen of 12:12 light: dark cycle. The mice were fed a diet from Ssniff (Soest, Germany) and had access to reverse osmosis water enriched with a mineral mixture ad libitum.

For the melanoma model, 5 million SK-MEL-37 tumor cells were subcutaneously implanted near the right scapula of the experimental animals suspended in 100 µL RPMI medium. The experiments strictly adhered to humane and ethical standards outlined in the European Community directive (86/609/EEC). Mice were closely monitored every 2–3 days for changes in skin condition, motor activity, and behavior. If mice displayed signs of toxicity (e.g., curvature, hunching, reduced activity), a body weight loss exceeding 20%, or a significant increase in tumor volume, they were euthanized in accordance with ethical guidelines for animal care. Planned euthanasia was carried out using CO_2_ overdose, followed by cervical dislocation. Tumor volumes were precisely determined using caliper measurements and the formula V = a × b2 × 0.52 (where ‘a’ represents length and ‘b’ represents width). When the average tumor volume reached 100 mm^3^, the mice were randomly distributed into three groups. The control group received 8 million non-transduced T-cells intravenously, the experimental group received anti-MAGEA4 TCR-like CAR-T cells in the same manner, and the third group remained untreated. Statistical analysis was conducted using GraphPad Prism 10.0.0 software (GraphPad Software, USA). The two-way ANOVA test was used to compare between the groups. Tumor volumes are presented as mean ± standard error of the mean.

## 5. Conclusions

In the course of this investigation, we conducted comprehensive in vitro assessments to evaluate the cytotoxic activity of MAGE-A4-specific CAR-T lymphocytes against the SK-Mle-37 tumor cell line expressing the target MAGE-A4 antigen. Our diverse in vitro functional assays not only corroborated tumor-specific cytotoxicity but also elucidated the presence of early activation markers, contributing to augmented antitumor activity. The observed enhancement in cytotoxic activity of the genetically modified T-lymphocytes can be attributed to the inclusion of GITR in the CAR receptor construct. Our initial in vivo experiments demonstrated a significant deceleration in tumor growth in a murine melanoma model. These findings underscore the potential of targeted immunotherapies for solid tumors and provide a valuable foundation for further research and clinical development. As we move forward, optimizing dosing regimens and conducting rigorous safety assessments will be crucial in advancing this promising therapeutic approach, while additional investigations into potential off-target effects and long-term efficacy will be pivotal for a more comprehensive understanding of its potential.

## Data Availability

The datasets generated and analyzed during the current research are accessible from the corresponding author upon an email request.

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
