# Peer review of "Exploring TCR-like CAR-Engineered Lymphocyte Cytotoxicity against MAGE-A4"

_ijms, 2023, doi:10.3390/ijms242015134_

Round 1
Reviewer 1 Report
Article "Exploring TCR-like CAR-Engineered Lymphocyte Cytotoxicity Against MAGE-A4 In Vitro." (Authors: Alaa Alsalloum and at. all). This paper describes promising candidate for targeted antitumor therapy with low toxicity. Therapeutic agent - TCR-like chimeric antigen receptor (CAR-T); target on the surface of cancer cells - tumor antigen MAGE-A4 (NW-Mel-38 and SK-Mle-37 tumor cell line). MAGE-A4 is a cancer-embryonic antigen that is primarily expressed in immune-privileged tissues such as the testicles and placenta. Authors assumed that the inclusion of the GITR activation motif within the intracellular domain of the CAR construct approach holds the potential to improve its therapeutic efficacy. This paper describes the generation, as well as phenotypic, and functional properties of lymphocytes with a TCR-like CAR receptor, the antigen-recognizing region of which is specific to the p230-239 epitope (GVYDGREHTV) of the tumor antigen MAGE-A4, and the intracellular domain contains the activation motif GITR (glucocorticoid-induced TNFR-related protein, TNFRSF18, CD357). The article corresponds to the subject of the journal " ijms". The article is well structured, written in a clear and understandable language, the conclusions are logical, the literature corresponds to the stated topic. The main conclusion of the study is that when genetic modification of lymphocytes active expressed markers CD69, CD107a, and FasL upon encountering tumor cells, indicating robust T-cell activation and cytotoxicity. As such, there are no comments on the work, however, I would like to see more than two cell lines in the work.
I recommend publishing your work.
Author Response
Dear reviewer 1, We would like to thank you for taking the necessary time and effort to revise the manuscript. We sincerely appreciate your positive comments about the manuscript.

Reviewer 2 Report
The authors mention immune transcriptome profiling and the upregulation of T-effector genes in TCR-like CAR T-cells. It is important to provide more specific information about the genes that were analyzed and their functional relevance. Additionally, the authors should provide insights into the overall immune response elicited by TCR-like CAR T-cells, including the activation of other immune cell types and potential immunomodulatory effects.
The authors mention increased cytokine production by TCR-like CAR T-cells, driven by granzymes and soluble Fas ligand (sFasL). To support this claim, the authors should provide quantitative data on the levels of these cytokines and compare them with control groups. Additionally, it would be valuable to explore the expression of other relevant cytokines involved in the anti-tumor immune response.
While the findings from in vitro experiments are promising, the authors acknowledge the need for in vivo investigations to fully validate their results. It is important for the authors to discuss the potential limitations of the in vitro study and outline their plans for future in vivo experiments. This could include animal models, tumor xenograft studies, or other relevant approaches.
Author Response
Dear reviewer 2, We would like to thank you for taking the necessary time and effort to revise the manuscript. We sincerely appreciate all the valuable comments and suggestions, which helped us improve the quality of the work. All of the recommendations have been addressed in the manuscript. Revised parts are marked up using the "Track Changes" function.
Point 1: The authors mention immune transcriptome profiling and the upregulation of T-effector genes in TCR-like CAR T-cells. It is important to provide more specific information about the genes that were analyzed and their functional relevance. Additionally, the authors should provide insights into the overall immune response elicited by TCR-like CAR T-cells, including the activation of other immune cell types and potential immunomodulatory effects.
Response 1: In response to your feedback, we've enriched our discussion by including detailed gene information and shedding light on the broader immune response induced by TCR-like CAR T-cells, encompassing their influence on various immune cell types and potential immunomodulatory effects. These updates align with your valuable input to enhance the depth of our research discussion.
Point 2: The authors mention increased cytokine production by TCR-like CAR T-cells, driven by granzymes and soluble Fas ligand (sFasL). To support this claim, the authors should provide quantitative data on the levels of these cytokines and compare them with control groups. Additionally, it would be valuable to explore the expression of other relevant cytokines involved in the anti-tumor immune response.
Response 2: In our experiment, we utilized the LEGENDplex™ Human CD8/NK Panel (13-plex) to analyze a specific set of 13 cytokines. As requested, we have now included a new table displaying the concentration levels of these cytokines. The data is presented as median values along with the interquartile range for clarity. Additionally, we have replaced the previous figure with an updated version that provides data for IL-2, IL-4, and IL-10. Furthermore, we have clarified in the Methods section that, as a control, we used cells primed with anti-CD3 and stimulated with IL-2, to which no retrovirus was added, and no transduction was performed. These enhancements offer comprehensive quantitative insights into cytokine production and further bolster the evidence supporting our findings regarding the impact of TCR-like CAR T-cells on cytokine levels and their potential role in the anti-tumor immune response.
Point 3: While the findings from in vitro experiments are promising, the authors acknowledge the need for in vivo investigations to fully validate their results. It is important for the authors to discuss the potential limitations of the in vitro study and outline their plans for future in vivo experiments. This could include animal models, tumor xenograft studies, or other relevant approaches.
Response 3: Thank you for your recommendation, and we appreciate your valuable input. In our revised discussion, we have addressed the potential limitations of our in vitro study, emphasizing the need for in vivo studies to bridge the gap between preclinical research and clinical translation. Additionally, we have outlined our plans for future in vivo experiments, which will involve animal models and tumor xenograft studies to further validate our findings.

Round 2
Reviewer 2 Report
The Authors addressed all the points and it sounds good to me to publish.
Author Response
Dear reviewer 2, We would like to thank you for taking the necessary time and effort to revise the manuscript. We sincerely appreciate your positive comments about the manuscript.
